Will the California Current lose its nesting Tufted Puffins?

Hart Christopher J. chrish32@uw.edu 1
Kelly Ryan P. 1
Pearson Scott F. 2
1 School of Marine and Environmental Affairs, University of Washington , Seattle , WA , United States of America
2 Washington Department of Fish and Wildlife , Olympia , WA , United States of America
Santoso Agus
Electronic publication date: 2018 Mar 22
Publication date: 2018
Volume: 6
Electronic Location ID: e4519
Received 2017 Nov 29; Accepted 2018 Feb 28
Copyright: ©2018 Hart et al.
Copyright year: 2018
Copyright holder: Hart et al.
License: This is an open access article distributed under the terms of the Creative Commons Attribution License, which permits unrestricted use, distribution, reproduction and adaptation in any medium and for any purpose provided that it is properly attributed. For attribution, the original author(s), title, publication source (PeerJ) and either DOI or URL of the article must be cited.
License URL: https://creativecommons.org/licenses/by/4.0/

Keywords: Endangered species act, Conservation biology, Climate change, Tufted puffin, Endangered species management, Habitat loss, Species distribution model

Funding: The authors received no funding for this work.

==============================
Tufted Puffin (Fratercula cirrhata) populations have experienced dramatic declines since the mid-19th century along the southern portion of the species range, leading citizen groups to petition the United States Fish and Wildlife Service (USFWS) to list the species as endangered in the contiguous US. While there remains no consensus on the mechanisms driving these trends, population decreases in the California Current Large Marine Ecosystem suggest climate-related factors, and in particular the indirect influence of sea-surface temperature on puffin prey. Here, we use three species distribution models (SDMs) to evaluate projected shifts in habitat suitable for Tufted Puffin nesting for the year 2050 under two future Intergovernmental Panel on Climate Change (IPCC) emission scenarios. Ensemble model results indicate warming marine and terrestrial temperatures play a key role in the loss of suitable Tufted Puffin nesting conditions in the California Current under both business-as-usual (RCP 8.5) and moderated (RCP 4.5) carbon emission scenarios, and in particular, that mean summer sea-surface temperatures greater than 15 °C are likely to make habitat unsuitable for breeding. Under both emission scenarios, ensemble model results suggest that more than 92% of currently suitable nesting habitat in the California Current is likely to become unsuitable. Moreover, the models suggest a net loss of greater than 21% of suitable nesting sites throughout the entire North American range of the Tufted Puffin, regardless of emission-reduction strategies. These model results highlight continued Tufted Puffin declines—particularly among southern breeding colonies—and indicate a significant risk of near-term extirpation in the California Current Large Marine Ecosystem.

Introduction

Worldwide, species are facing increasing challenges associated with rising sea- and air surface temperatures (Thomas et al., 2004). Warming climates have resulted in distribution and community-abundance changes in species ranges across multiple taxa (Parmesan & Yohe, 2003), changes to ecological responses including phenological anomalies correlating with warming seasonal temperatures (Walther et al., 2002), and changes in habitat quality and distribution (Klausmeyer & Shaw, 2009). Foden et al. (2013) found that 83% of birds, 66% of amphibians and 70% of corals that were identified as highly vulnerable to the impacts of climate change are not currently considered threatened with extinction on the IUCN Red List of Threatened Species, indicating that species’ vulnerabilities are likely to be much greater than conservation status alone would suggest.

In recent years in the United States, the United States Fish and Wildlife Service (USFWS) and the National Marine Fisheries Service (NMFS) have received several petitions to list species under the Endangered Species Act due to the impacts of climate change (Siegel & Cummings, 2005; Wolf, Cummings & Siegel, 2008). However, the link between climate change and risk to a species can be difficult to assess. One approach to examining these linkages is to model the interaction between climate and suitable habitat for a given species, given what is already known about the relationship between the species and its habitat. This approach has become an integral component of conservation planning in a world of changing environments (Hagen & Hodges, 2006; Richardson & Whittaker, 2010). Ultimately, understanding these linkages can help inform conservation assessments and species and ecosystem management strategies (Carnaval & Moritz, 2008; Ponce-Reyes et al., 2017), for example, by estimating the likelihood of losing (or gaining) particular suitable habitats of interest under future climate conditions.

Tufted puffins as a pertinent example

The Tufted Puffin (Fratercula cirrhata) is an iconic species that is experiencing dramatic population declines across the southern portion of its geographic range (Piatt & Kitaysky, 2002). While Tufted Puffin populations in the Alaska Current have remained relatively stable (but see Goyert et al., 2017), populations in the California Current large marine ecosystem (area of the eastern Pacific Ocean spanning nearly 3,000 km from southern British Columbia, Canada to Baja California, Mexico) have declined by approximately 90% relative to early 20th century estimates, and are currently declining 9% annually (Hanson & Wiles, 2015). The number of occupied breeding-colony sites in Washington State has declined by 60% relative to the 1886–1977 average, and 45% relative to the 1978–1984 average (Hanson & Wiles, 2015). Range contractions at the southern edge of the Tufted Puffin’s habitat in both the eastern and western Pacific Ocean have led to preliminary conservation measures: the Washington Department of Fish and Wildlife has listed the Tufted Puffin as endangered at the state level in 2015 with Japan’s Ministry of the Environment listing the species as endangered in 1993 (Osa & Watanuki, 2002; Hanson & Wiles, 2015; Washington Fish and Wildlife Commission, 2015).

Tufted puffin biology and ecology

Tufted Puffins are seabirds belonging to the family Alcidae and nest in colonies located on both sides of the North Pacific, ranging in North America from the Channel Islands in southern California (34°N) to coastal northern Alaska (68°N) (Piatt & Kitaysky, 2002) and in Asia from Hokkaido, Japan (43°N) through the Kamchatka Peninsula (63°N) (Hanson & Wiles, 2015). They are central-place foragers during the nesting season, when they dig burrows or use crevices for nesting on nearshore rocks, islands and sea stacks (Piatt & Kitaysky, 2002). During the nesting season, puffins exhibit large foraging radii around their colonies (up to 40 km, e.g., Menza et al., 2016: Fig. 12) and are able to carry more than twenty fish at a time while flying back to the colony to feed their chicks (Piatt & Kitaysky, 2002; Hanson & Wiles, 2015). While little is known about the wintering distribution and ecology of Tufted Puffins, summer (May–September) breeding colonies are well documented and provide the most useful biological data for conservation management (Piatt & Kitaysky, 2002). Extensive breeding colony surveys dating back to the early 20th century allow us to examine any potential link between climate and species range extent.

Tufted Puffins are subject to multiple well-documented ecological stressors—such as increasing eagle predation, habitat degradation, declining prey availability, and fishing net entanglement (Baird, 1991; DeGange & Day, 1991; Ricca, Keith Miles & Anthony, 2008)—but several mechanisms associated with temperature stress may be important in driving puffin declines along their southern range boundary. Gjerdrum et al. (2003) found dramatically reduced growth rates and fledging success (development of fledgling wings and muscles for flight) in years with high sea surface temperature (SST) anomalies. Other researchers cite the nutritional demands of puffin chicks and the prey availability and preferences correlating with fledgling success (Hipfner, Charette & Blackburn, 2007), suggesting a mechanism for the negative effects of high sea surface temperature on puffin chicks. These and other studies point to a link between temperature and demographic trends in the Tufted Puffin and help identify this species as a candidate for distribution modeling. Modeling outputs may help expose proposed interactions between high ocean temperatures, prey distribution in the water column and puffin breeding success.

As a result of these potential threats and documented population declines, the USFWS was petitioned to list the California Current population of the Tufted Puffin (Fratercula cirrhata) as endangered under the Endangered Species Act (ESA) (Sewell, 2014). In order to respond to this petition, the USFWS is currently examining Tufted Puffin status and trends, evaluating threats to its survival, the adequacy of existing regulatory mechanisms to conserve the species, the loss of its habitat, and other relevant factors. Given that climate—specifically, increasing sea-surface temperatures—may be a particularly important factor influencing puffin population dynamics and ultimately reducing puffin breeding range, and given the vast geographic extent of puffin nesting sites (34° of latitude and roughly 70° of longitude in North America) and historical data on the occupancy of these sites, the Tufted Puffin is an excellent candidate for species distribution modeling.

Species distribution models in conservation planning

Species distribution models (SDMs) are a powerful way to examine how climate variables relate to species geographic distribution and the distribution of suitable habitat (Guisan & Zimmermann, 2000; Guisan & Thuiller, 2005). By associating species occurrence with climate variables, these models can: (1) test for associations in space and time between putative environmental drivers and changes in species range and (2) project changes in suitable habitat under future climate change scenarios (Bellard et al., 2012). SDMs use a variety of underlying statistical models to capture the relationship between habitat and climate and create detailed outputs highly useful for wildlife management (Carvalho et al., 2011; Guisan et al., 2013). Recent Endangered Species Act listing decisions and management plans have drawn on SDM results to provide critical spatial and temporal conservation information. For example, climate envelope models were used to develop spatially explicit conservation strategies that account for climate change, notably in the case of the North America Wolverine (Gulo gulo), where the models were the basis of an ESA listing (United States Fish and Wildlife Service, 2016).

Here we use 50 years of nesting-habitat distribution information—ranging from the Bering Sea to California—to map Tufted Puffin nesting habitat. We use this colony occupancy data to model the relationship between nesting habitat and current environmental conditions to project future suitable breeding sites in the same geographic range. We present these results as an example of how this information can be used in both regulatory (e.g., Endangered Species Act) and conservation planning contexts.

Materials and Methods

Environmental data

Environmental data for the current period, which we define here as the years 1950–2000, was downloaded from WorldClim, a set of global climate layers derived from interpolation of monthly climate observations (Hijmans et al., 2005, last accessed January 2017). After removing WorldClim bioclimatic variables displaying high collinearity, we also considered factors relevant to Tufted Puffin breeding phenology including isolating seasons during which puffins breed (spring and summer) and including environmental variables relevant to their prey species such as temperature. These processes resulted in our selection of six environmental variables for analysis: annual temperature range (ATR), mean diurnal temperature range (MDR), mean temperature of the warmest quarter (MTWQ), annual precipitation (AP), precipitation of the warmest quarter (PWQ) and distance-to-ocean (DIST), a variable we created to help models discern suitable nesting habitat as occurring only in rocky, coastal habitats within meters of the sea, a biological requirement of puffins (Piatt & Kitaysky, 2002); (see Table S1 for measurements and units). Each variable for the current period was scaled to a 5 arcmin grid cell size (ca 10 km × 10 km). After scaling, all environmental variables within the relevant geographic range were cropped to only include locations within 200 kilometers of the ocean. While this cropping distance includes land not physically suitable for obligate coastal and island breeders, it is important for creating larger environmental variable gradients during model construction (Van Horn, 2002).

The same six environmental variables above were averaged over the period of 1910-1950 to construct a ‘past’ climate regime used to project past Tufted Puffin range. Past climate variables were selected using gridded climate data obtained from monthly observations from the Climate Research Unit CRU TS v. 4.01 dataset (Harris et al., 2014 (crudata.uea.ac.uk), last accessed March 2017). Past environmental data was similarly scaled down to the same 5 arcmin grid cell size as the current data.

Future climate

We selected emissions scenarios Radiative Concentration Pathways (RCP) 4.5 and 8.5 as defined by the IPCC 5th Assessment Report (IPCC, 2014) as future environmental projections against which to forecast changes in Tufted Puffin breeding distribution. Downscaled model output for environmental variables for both future RCP scenarios were averaged across the following general circulation models: Hadley Centre’s HadGEM2-AO (Collins et al., 2011), NOAA’s GFDL-CM3 (Griffies et al., 2011), NASA’s GISS-E2-R (Nazarenko et al., 2015), Institut Pierre-Simon Laplace’s IPSL-CM5A-LR (Dufresne et al., 2013), Beijing Climate Center’s BCC-CSM1-1 (Xin, Wu & Zhang, 2013), Bjerknes Centre’s NorESM1-M (Bentsen et al., 2013), National Center for Atmospheric Research’s CCSM4 (Gent et al., 2011) and the Max Planck Institute’s MPI-ESM-LR (Giorgetta et al., 2013), all for the year 2050 (average of 2041–2060) (Hijmans et al., 2005, last accessed January 2017). Using the average of eight prominent climate model outputs helps incorporate variance in potential future climate projections within our model. The 2050 timeframe and these emissions scenarios (roughly speaking, a moderate-reduction scenario and business-as-usual scenario with no emission reductions) were selected as the most relevant to the conservation decisions presently surrounding the Tufted Puffin (IPCC, 2014).

Species data

Species distribution data were obtained courtesy of USFWS, Washington Department of Fish and Wildlife and Environment and Climate Change Canada, and were derived from expansive US and Canada breeding colony surveys conducted by groups including USFWS, Washington Department of Fish and Wildlife, Alaska Department of Fish and Game, Environment and Climate Change Canada, California Department of Fish and Wildlife, and others (Speich & Wahl, 1989; Hodum et al., see Supplemental Information 4; World Seabird Union (https://seabirds.net, last accessed March 2017); British Columbia Marine Conservation Analysis (http://bcmca.ca/datafeatures/eco_birds_tuftedpuffincolonies/), 2017, last accessed May 2017), see Supplemental Information 5). Count data consisted of estimates of numbers of breeding individuals present at known nesting colonies and the spatial coordinates of those observations. Biological data for the ‘current’ period of climate data (see Table S2) represents the most recent survey observation of known nesting sites from 1950–2009. While the climatological data runs until the year 2000, biological data from up to 2009 was included to incorporate recent detailed state-wide surveys in both Oregon and Washington, information critical to examining trends across the puffin’s southern range. We converted count data to presence/absence values, given the nature of our analysis, which asked whether breeding habitat was likely to be suitable (≥1 nesting birds) or not (0) under future conditions. Some observations were adjusted geographically up to one grid cell (ca 10 km) to fall within gridded terrestrial environmental variables. Observations further than 15 km from terrestrial grids (e.g., remote islands) were removed from the analysis. The environmental variables described above were selected to model potential interactions between climate conditions and puffin range during the breeding season.

Given the low proportion of absence-to-presence observations for Tufted Puffin surveys and potential bias in survey locations, we added pseudo-absence (PA) observations (i.e., generated absence observations existing within the range of the SDM) to all models. SDMs using both presence and absence have been shown to perform more accurately than models relying on presence-only observations (Elith et al., 2006; Barbet-Massin et al., 2012). PA generation methodology is also important in both model predictive accuracy and avoiding model over-fitting (Barbet-Massin, Thuiller & Jiguet, 2010; Barbet-Massin et al., 2012). Adapting these recommendations in Barbet-Massin et al. (2012) 1000 PAs were randomly generated twice across the SDM a minimum of 30 km from any presence or true absence point.

Model parameterization

Because Tufted Puffins rely heavily on both terrestrial and marine environments for reproduction, we initially tested the correlation between sea-surface temperature and air-temperature data across puffin colony observations. SST data for this comparison comprised an average of mean monthly temperature for June, July and August, months aligning with Tufted Puffin breeding season obtained from the Hadley Centre, UK (Rayner et al., 2003 (metoffice.gov.uk/hadobs) last accessed March 2017) and the corresponding air temperature readings (MTWQ) (Hijmans et al., 2005). Both sets of environmental variables were scaled to a 5 arcmin grid cell size and represented means from the years 1950–2000. A high correlation coefficient (r = 0.96) allowed us to use air temperature—which is available in higher spatial resolution—rather than SST in the final analysis. This strong relationship between air- and sea-surface temperature has also been documented across several other marine and aquatic species distribution studies (Stefan & Preud’homme, 1993; Domisch et al., 2013). Additionally, within R software (R Core Team, 2013), a principal component analysis (PCA) (Pearson, 1901) was performed to compare variance in environmental variables between areas of collapsed colonies (absence) and occupied colonies during the current period. This technique can help identify differences in environmental niches of species occurrence data (e.g., Broennimann et al., 2012; Peña Gómez et al., 2014); here, we use it to create an index of environmental variables to identify likely drivers of Tufted Puffin declines after accounting for the covariances among variables.

Species distribution modeling

Model algorithms

SDMs were constructed with the R package BIOMOD2 (BIOdiversity MODelling) (Thuiller et al., 2009; R Core Team, 2013). All SDMs were constructed for a spatial range larger than the current estimated US Tufted Puffin distribution (180°W to 120°W longitude and 33°N to 69°N). Using a larger extent both increases the range of environmental gradients available for model construction and introduces novel climates useful for projecting potential migration (Thuiller et al., 2004; Fitzpatrick & Hargrove, 2009; Domisch et al., 2013). Models were also examined under a subset of all biological and environmental data from 126°W to 120°W and 32°N to 48.5°N. This portion of the analysis is intended to account for the spatial variance of puffin distribution and examine the temperature-habitat relationship in the California Current large marine ecosystem exclusively—the portion of the range that has experienced the greatest decline and has been petitioned for listing under the US Endangered Species Act.

To help acknowledge and estimate uncertainty, three different models using different statistical approaches were selected from the BIOMOD framework; generalized linear models (GLM) (McCullagh & Nelder, 1989), generalized boosting models (GBM, also referred to as boosted regression trees) (Ridgeway, 1999) and random forests (RF) (Breiman, 2001). The GLM models used a logit link between the response variable mean and combination of explanatory variables (Guisan, Edwards & Hastie, 2002) (i.e., logistic regression). GBMs incorporate regression and machine-learning techniques through boosting many decision-tree models to increase model performance (Elith, Leathwick & Hastie, 2008). Decision models recursively partition sets of explanatory and outcome variables in a stagewise manner until subsets of data are explained by trees of bifurcating decisions (Elith, Leathwick & Hastie, 2008). Boosting then sequentially fits decision trees to training data, selecting the trees that best fit the data (Elith, Leathwick & Hastie, 2008). Finally, RF is a machine-learning technique that creates classification trees similar to those in GBMs, but instead uses random bootstrap samples of data and explanatory variables upon the construction of each tree (Breiman, 2001).

The differences in statistical and machine learning approaches across GLM, GBM and RF algorithms provides variance across which to test sensitivity between models as well as estimations of model uncertainty (Marmion et al., 2009; Rodríguez-Castañeda et al., 2012). Additionally, using models with relatively more ensemble (GBM and RF) and parsimonious (GLM) approaches to habitat selection as well as utilizing both parametric (GLM) and non-parametric (RF) techniques provides robust analysis of environmental drivers of range change (Marmion et al., 2009) and led to the selection of these three model algorithms.

Model calibration

Having generated two variants of the dataset by generating distinct pseudo-absences, we then constructed twenty models for each algorithm (GLM, GBM, RF), for each dataset variant, for a total of 120 models. All models then used past environmental data as well as future emission scenarios to project both past and future puffin range changes. Each model variant performed a random 70:30 split of the biological data using 70% for model calibration and 30% for model evaluation. This technique addresses spatial autocorrelation and is frequently utilized when faced with dependent biological sampling (surveying of species around only areas of known occurrence) (Araújo et al., 2005). Model selection and calibration parameters were kept constant between past and current models to maintain consistency and repeatability. For all models across all algorithms, the default model construction options and parameters of the BIOMOD package were used (Thuiller et al., 2009).

Hindcasting

Hindcast models were created to examine the past relationship between temperature and patterns of puffin colony occupancy. Hindcasts can increase confidence in future projections and help shed light on ecological interactions over time (Raxworthy et al., 2003; Labay et al., 2011). These ‘hindcast’ models use current puffin distribution projections and the past environmental data detailed above to project past puffin colony distribution. A limited amount of historical survey data along the California Current, especially in southern Oregon and California, makes scoring the hindcast projections against past survey data difficult for this analysis. However, examining past model projections at specific known historical locations (such those in Hanson & Wiles, 2015) in a past, cooler climate helps interpret the influence of a warming climate in the future (Labay et al., 2011).

Ensemble modeling and evaluation

The area under the receiver operating characteristic curve (AUC) and the True Skill Statistic (TSS) were the two model metrics used to evaluate model performance. AUC maps sensitivity rate (true positive) against (1-specificity) values (=false-positive rate) and is a popular metric for species distribution model evaluations because it evaluates across all thresholds of probability conversion to binary presence or absence (Fielding & Bell, 1997; Guo et al., 2015). Higher AUC scores represent better model performance, with AUC scores between 0.7–0.8 classified as ‘fair’, 0.8–0.9 as ‘good’ and 0.9–1.0 as ‘excellent’ (Guo et al., 2015). TSS scores display (sensitivity + specificity −1) with sensitivity quantifying omission errors and specificity quantifying commission errors (Allouche, Tsoar & Kadmon, 2006; Guo et al., 2015; Shabani, Kumar & Ahmadi, 2016). TSS scores of zero or less indicate model performance no better than random and scores of 1.0 indicating perfect performance. Both scores were emphasized in this analysis to provide strong measures of ordinal model performance and to predict accuracy in threshold-dependent conservation planning (Allouche, Tsoar & Kadmon, 2006; Shabani, Kumar & Ahmadi, 2016).

Ensemble models were created using weighted averages of TSS scores both within and across algorithms while AUC scores were not used in constructing ensemble models but served as another evaluation metric. This technique captures uncertainty stemming from random sampling of the dataset as well as variance across modeling techniques (Gallardo & Aldridge, 2013), thereby providing the user with a robust sense of model fit and sensitivity to particular parameters. TSS scores below 0.7 were excluded from the ensemble to remove influence from poor predictive models (Araújo et al., 2011). A proportional weight decay was used averaging model weights, resulting in weights proportional to TSS evaluation scores. Additionally, binary conversions, which maximized model TSS performance, were used in some range-change analyses. Range-change analyses were performed allowing future migration to potential suitable future habitat as well as with no potential migration. Ensemble binary thresholds and their impact on projections are noted below.

Results

Model performance

Models from all three algorithms, and especially the ensemble model, scored very high in both model performance metrics (Table 1). GLM, GBM and RF algorithms displayed mean TSS scores and standard deviations of 0.842 ± 0.020, 0.898 ± 0.017 and 0.905 ± 0.017, respectively. Similarly, GLM, GBM and RF mean AUC scores were very high, indicating good model accuracy (Table 1). Techniques using machine learning methods (RF and GBM) consistently displayed the highest performance by both AUC and TSS scores (Table 1), perhaps due to these machine-learning (GBM and RF) models relying on boosting and ensemble learning, respectively, compared to a single regression model approach within GLM algorithms. Despite the different statistical and learning approaches of the selected modeling approaches, TSS and AUC scores were high across all techniques and displayed low variance (Table 1).

Table 1 Evaluation metrics and range change analysis for ensemble model and by model algorithm (N = 40).

(A) Model area under the receiver operating characteristic curve (AUC) and true skill statistic (TSS) for ensemble model and by algorithm. AUC represents sensitivity rate (true positive) against 1-specificity values (false positive) and TSS represents (sensitivity + specificity −1). Scores presented are mean plus or minus standard deviation (B) Percent of projected change in range by model algorithm. North America-wide and US California Current (32°N–48.5°N) independent analyses. Both RCP 4.5 (4.5 also italicized) and RCP 8.5 (8.5 also bolded) represented. Scores presented are mean plus or minus standard deviation.

	Ensemble	GLM	GBM	RF	
(A) Model evaluation					
TSS	.920	.842 ± .020	.898 ± .017	.905 ± .017	
AUC	.994	.976 ± .004	.985 ± .004	.986 ± .004	
(B) % Range change					
4.5         Species-wide	−21.80	−22.59 ± 11.69	−13.88 ± 11.72	−17.99 ± 20.33	
California current	−92.68	−96.69 ± 14.61	−16.95 ± 20.12	−21.26 ± 26.10	
8.5         Species-wide	−26.14	−31.37 ± 14.70	−14.09 ± 13.74	−19.23 ± 23.79	
California current	−97.56	−97.13 ± 12.96	−22.71 ± 20.27	−27.26 ± 26.80	

Variable contribution

Response plots

After initial variable winnowing, both model response plots and PCA analysis indicate that temperature variables ATR and MTWQ are strongly associated with Tufted Puffin breeding habitat (Figs. 1, 2). Importantly, MTWQ displayed a thermal maximum of suitable nesting habitat (i.e., a threshold) around 15 °C, and the MTWQ variable also displayed the most consensus across model members among all selected variables (Fig. 1). This result highlights consensus among GLM ensemble members surrounding the 15 °C threshold. ATR and MTWQ are related to extreme summer temperatures and consensus among GLM ensemble members across these variable response plots is consistent with the hypothesis that summer temperature anomalies influence Tufted Puffin colonies.

Figure 1 General Linearized Model (GLM) model algorithm variable response plots.

Response curves across GLM algorithms for all environmental variables. (A) Annual Temperature Range (ATR) response curves (B) Mean Diurnal Temperature Range (MDR) response curves (C) Mean Temperature of the Warmest Quarter (MTWQ) response curves (D) Annual Precipitation (AP) response curves (E) Precipitation of the Warmest Quarter (PWQ) response curves (F) Distance from Coast (DIST) response curves. Each colored line represents one GLM model run (N = 40). Y-axis displays predicted probability of habitat suitability across each variable given other variables are held fixed at their mean value. X-axis displays environmental variable values (see Table S1 for specific units). Results display distinct cutoffs between ATR, MTWQ and occurrence probability.

Conversely, MDR and PWQ were not effective in predicting suitable habitat among all models. In fact, the probability of Tufted Puffin occurrence remains high across the range of MDR and PWQ values indicating that these variables are not helpful in predicting puffin occupancy. GLM models do show a response to increased annual precipitation (AP) values, but there remains a lack of consensus among model members around a particular response cutoff.

Principal component analysis

Principal component analysis provided further support to the hypothesis identifying summer temperature as a primary driver of variance in Tufted Puffin breeding habitat (Fig. 2). PCA components 1 (51%) and 2 (27%) together explained 78% of the variability in the data. Component 2, with a strong loading of MTWQ of −0.732 and MDR loading of −0.497, indicates that MTWQ and MDR explain the difference between presence and absence points as evidenced by the separation of the 95% confidence ellipse along this component (Fig. 2). The other four variables loaded more strongly onto principal component 1 which does not help separate presence and absence points. This result combined with the MTWQ response curves (Fig. 2) indicate the importance of MTWQ in predicting what habitat is suitable for Tufted Puffins (Figs. 1 and 2).

Figure 2 Principal component analysis loadings 1 and 2 (95% confidence ellipses) for occupied (present) and unoccupied (absent) nesting colonies.

Range forecasts

North American projections—2050

After binary transformation of the future probabilistic projection maps, ensemble models, again representative of the weighted mean of all model algorithms and variants, project a range loss of approximately 22% of currently occupied range under RCP 4.5 and a range loss of approximately 26% under RCP 8.5 by 2050 (Table 1). GLM models projected the greatest percent habitat loss across North America under both emission scenarios (Table 1). There was uniform agreement across ensemble algorithms in projecting habitat loss, even with the possibility of colonizing new habitat, with variability among algorithms as to the magnitude of that loss (Table 1). Spatially, losses were uniformly projected along the California Current up to southeastern Alaska (Fig. 3), although ensemble projections suggested continued suitability of the Aleutian Islands under both emission scenarios (Fig. 3). Ensemble model results also reflected agreement on the opportunity for northward range expansion (Fig. 3). Both the projected southern range contraction and northward range expansion are further consistent with the hypothesized relationship between puffin habitat and temperature.

Figure 3 North-America-wide habitat projection maps.

Tufted Puffin breeding habitat range projection maps. Probabilistic maps, color bins display percent probability of grid cell representing suitable habitat. (A) Current projections. (B) 2050 projections under RCP 4.5. (C) 2050 projections under RCP 8.5. Map data ©2017 Google.

California current—2050

Analysis of the California Current region within the overall ensemble models shows near complete loss of suitable habitat between emission scenarios with both RCP 4.5 and 8.5 (Fig. 4), although the individual component models showed variable amounts of habitat loss. GLM models projected the most dramatic loss along the California Current with a predicted loss of greater than 96% of suitable habitat under both scenarios (SD = 14.61% for projections under RCP 4.5, SD = 12.96% for projections under RCP 8.5) (Table 1). GLM models also projected no habitat as likely to become newly habitable in the California Current under either emission scenario. Both GBM and RF models predicted less range change with GBM models projecting a mean loss of approximately 23% across algorithm variants and RF models projecting a mean loss of approximately 27% across algorithm variants under RCP 8.5 (SD = 20.27%, SD = 26.80%, respectively). Under both RCP 4.5 and 8.5, ensemble projections display complete loss of likely suitable habitat in Oregon and virtually complete loss in California (Fig. 4). GBM and RF models projected small portions of northwestern Washington would become slightly more likely than not to become environmentally habitable by 2050 under both emission scenarios, though those locations may not exhibit other puffin habitat requirements.

Figure 4 California Current habitat projection maps.

Tufted Puffin breeding habitat range projection maps exclusive to the California Current (32°N–48.5°N). Probabilistic maps, color bins display percent probability of grid cell representing suitable habitat. (A) Current projections. (B) 2050 projections under RCP 4.5. (C) 2050 projections under RCP 8.5. Note: Probability bin “61–78%” absent in B and C as projections do not reflect any habitat within that bin. Map tiles ©Stamen Design, underCC BY 3.0. Data ©OpenStreetMap, underODbL.

Hindcast

As stated above, limitations on biological survey for the Tufted Puffin make interpretation of hindcast results difficult, though past projections further supported the hypothesis of higher temperature limiting suitable nesting habitat. See supplementary material for further discussion and presentation of hindcast model results.

Discussion

Ensemble models uniformly support summer temperature as a predictor of Tufted Puffin breeding habitat. High model evaluation metrics (Table 1) coupled with strong correlations between temperature variables and Tufted Puffin range change (Figs. 1 and 2) provide confidence that projected warmer summer temperatures are likely to be associated with the loss of greater than 92% of Tufted Puffin breeding habitat in the California Current under the examined emission scenarios (Fig. 4). North America-wide, ensemble models project an overall loss across model algorithms and variants of approximately 22% and approximately 26% of suitable habitat, respectively, under moderate emission reductions and business as usual carbon emissions by 2050 (Table 1). Figure 3 highlights that most nesting habitat will be lost along the southern portion of current Tufted Puffin range as well as the opportunity for northward range expansion.

Within the California Current, ensemble projections show little variance in the projected range loss under RCP 4.5 versus RCP 8.5. However, the model algorithms varied considerably with respect to the percent of future habitat likely to be suitable in more northerly latitudes (Table 1). This variance is likely due to the differences in modeling techniques across algorithms described above and variance in initial estimation of suitable habitat (Table 1, Fig. 4). Both ensemble models project minimal suitable habitat remaining along the California Current, therefore under the hypothesized relationship between temperature and suitable habitat, there is little left to become unsuitable with increased warming (Fig. 4). Another factor which may contribute to the lack of variance in projected habitat loss, range-wide, under different RCP scenarios in the California Current is the relatively short timeframe of 2050 projections. Divergence in the temperature projections of RCPs 4.5 and 8.5 are amplified after 2050 with less divergence in temperature in the short term (IPCC, 2014).

Important to the interpretation of ensemble projections is the binary transformation of model outputs into suitable and unsuitable categories. For range-change analyses, projections of unsuitable habitat represent a weighted average of <50% probability of suitability, a cutoff defined by ensemble calibration. In some cases we observed a majority of ensemble members projecting a particular cell as marginally suitable while a minority of members strongly project that cell as unsuitable. The subsequent result is unsuitable habitat despite being marginally suitable in some models. This process of binary transformation can then reflect an aggregate of probabilistic scores instead of the average of a binary projection. Binary transformations are thus a useful tool to discuss and represent how changes in climate may affect the likelihood of suitable breeding conditions throughout Tufted Puffin range, but are necessarily imprecise in that they mask underlying variability. Additionally, many sites in the ensemble projections (Fig. 4) become unsuitable after binary transformations while falling in to the 27%–44% probability bin. Though these are projected as likely to become environmentally unsuitable, they represent marginally suitable environmental habitat and further research may help determine the suitability of other requisite habitat conditions at these sites to provide a broader picture of nesting habitat suitability at these locations.

Examining the variance among model results and the spatial variance in projections is integral to the interpretation of model results from a conservation perspective (Guisan et al., 2013; Porfirio et al., 2014). Tufted Puffins are a relatively rare species in the southern portion of their range, are hard to survey, and occupy small areas of land (Hanson & Wiles, 2015). These biological factors contribute to the difficulty of surveying (and therefore modeling) puffins and can increase variance among model algorithms, making ensemble models more valuable for interpretation of results Segurado & Araujo, 2004; Hernandez et al., 2006. However, here we use colony occupancy information rather than counts. Preliminary occupancy analysis suggest that colony occupancy can be assessed with a high probability with a single relatively rapid visit by boat even to a very small colony with few birds (Pearson et al., see Supplemental File). Thus, our colony occupancy approach likely reflects actual changes in colony occupancy throughout the range. In addition, trends were consistent across algorithms in depicting significant losses of suitability for habitat across the California Current (specifically California and Oregon), British Columbia and eastern Alaska (Fig. 3). All algorithms also projected the opportunity for northward range expansion in the face of accelerating northern latitude warming (Fig. 3).

If suitable habitat expands northward as projected by our ensemble models, biological and ecological factors unrelated to climate such as eagle predation, requisite nesting substrate, etc., are predicted to continue and likely to influence the probability of colonization (Hipfner et al., 2012; Hanson & Wiles, 2015). Because colonization is uncertain, we depict in Fig. 5 the loss of currently suitable habitat in the California Current without the possibility of new colonization throughout the extent—a worst case scenario but an important component of analysis when examining the threat of climate change. Variance among models as evidenced in Table 1 along the California Current failed to result in more than a handful of consensus areas of suitability (Fig. 4). Ensemble models did depict some areas of marginally suitable habitat along central California (Fig. 4). This result was likely influenced by a few outlying colonies such as the one in the Farallon Islands, California. These outlying colonies persist in opposition to the trends seen in other colonies throughout the southern portion of puffin range. Further examination of the mechanisms driving puffin declines in their southern range may shed light on either the viability of these outlying suitable habitat projections as potential climate refugia or other mechanisms supporting these outlying colonies.

Figure 5 Histogram of habitat loss in the California Current with no migration.

Histogram displaying the amount of current California Current Extent suitable habitat projected to become unsuitable by 2050 under RCP 8.5 (N = 120). Colors represent model algorithms. In this analysis, there is an assumption of no migration or dispersal to potentially suitable new habitat.

The discussed hard-to-model specific requirements of puffin nesting habitat and other ecological population drivers make fine scale colony-by-colony analysis of extirpation risk difficult. Additionally, analysis of regional trends in puffin success can serve to guide research examining the specific causal mechanisms driving documented declines which would aid in further analysis of colony-by-colony extirpation risk. Importantly, all models and especially ensemble results support the trend of southern range contraction associated with warm summer temperatures (Figs. 1–4). Additionally, while limitations on historical survey data make interpretation of hindcasts difficult, preliminary hindcasting resulted in expansion across the southern portion of current puffin habitat (see Supplemental File). This result is further consistent with the hypothesized relationship between high temperature and puffin success.

Our results are especially salient in light of the ongoing US Fish and Wildlife Service’s analysis of puffin status following Natural Resources Defense Council’s petition to list the California Current population as endangered. When responding to the petition to list the puffin, the Service can list the species throughout its range or can list a distinct population segment (DPS) such as the breeding population south of the Canadian border or that in the California Current. While determining which segments comprise a DPS as outlined by the ESA requires more analysis, our results provide the spatial information to inform the threat that both of these breeding range segments or “populations” will likely face. Our results suggest all potential distinct populations segments from British Columbia, southward, face a significant chance of near extirpation or very significant habitat loss under a wide range of climate projections by 2050.

Conservation planning for species can greatly benefit from defining the portion(s) of their range representing habitat critical to their survival (Hagen & Hodges, 2006). This designation is essential for conservation planning both under the ESA as well as Canadian Species at Risk Act, in which it is required for listed species, as well as for more localized conservation efforts (Taylor, Suckling & Rachlinski, 2005). Figures 3 and 4 highlight areas where Tufted Puffins are currently at the highest risk of colony loss (low habitat suitability). Many puffin nesting sites are already managed by the US Fish and Wildlife Service refuge system and many of these sites are also designated as “wilderness” (Speich & Wahl, 1989 and United States Fish and Wildlife Service, 2017). Habitat projections made for the year 2050 permit analysis of critical habitat in terms of species survival as well as proposed conservation efficacy (Suckling & Taylor, 2006; Stein et al., 2013). Land acquisition has proven to be an effective strategy for the management of endangered species and is a strategy that has been utilized for the Tufted Puffin (Lawler, White & Master, 2003; WDFW, 2016) and could be used in the future. With limited resources to conserve species at the federal level, ranking the conservation priorities and temporally analyzing threats can allow for prudent investment in conservation lands (Lawler, White & Master, 2003). Nesting colony sites throughout the Gulf of Alaska are projected to remain suitable and results indicate the Aleutian Islands are the most likely habitat to both continue to support large populations of Tufted Puffins as well as potentially becoming suitable as new breeding sites (Fig. 3). As these results suggest, we can use this information to predict areas of future Tufted Puffin habitat to help outline areas for long-term conservation action while also mapping areas where long-term conservation efforts may prove ineffective. Such proactive conservation steps often result in greater conservation outcomes and are critical for species struggling to adapt to changing climates (Morrison et al., 2011).

Mechanisms driving decline

Using the results reflected in Figs. 3 and 4, wildlife managers can continue to explore the causal mechanisms driving the discussed Tufted Puffin population declines and range contraction. Currently numerous pathways are proposed to help determine puffin breeding success and adult survival such as prey availability, SST, predation and habitat degradation (Morrison et al., 2011; Hanson & Wiles, 2015). While many prey species do not show significant population trends (MacCall, 1996), our results can provide spatial details to explore a potential mechanistic explanation, vertical prey distribution (Gjerdrum et al., 2003). Exact measurements are unknown but based on body size, Tufted Puffins exhibit the deepest maximum forage depths across alcids, at approximately 110 m, but typically forage at 60 m or less (Piatt & Kitaysky, 2002). Tufted Puffins also forage much further offshore than most other alcids and in deeper waters along continental shelf breaks (Ostrand et al., 1998; Menza et al., 2016). Foraging in deeper waters may leave Tufted Puffins susceptible to downward movement of prey species in the water column during high temperatures (Ostrand et al., 1998; Gjerdrum et al., 2003). Further research around these biological and ecological factors can be combined with our model results to further explore the mechanisms behind the temperature-range relationship for Tufted Puffins (Ostrand et al., 1998; Piatt & Kitaysky, 2002).

In addition to uncovering causal mechanisms, current conservation efforts are beginning to examine diverging population patterns among related birds, Rhinoceros Auklets (Cerorhinca monocerata), Cassin’s Auklets (Ptychoramphus aleuticus) as well as Tufted Puffins along the California Current (Grémillet & Boulinier, 2009; Morrison et al., 2011). While these three alcids fill similar ecological roles, recent years have seen dramatic population swings varying among species (e.g., El-Niño of 1997–98) (Morrison et al., 2011). Cassin’s Auklets have displayed similar ecological sensitivity to changing environmental conditions and have experienced recent large scale mortality events as recently as 2015 (Sydeman et al., 2006; Wolf et al., 2010; Hanson & Wiles, 2015). Physiological and ecological differences between these related seabird species such as forage radius, foraging depth, and diet composition may provide insights into the mechanisms responsible for these differences in population trends among species (Sydeman et al., 2001; Wolf et al., 2009; Wolf et al., 2010; Morrison et al., 2011). For example, using SDMs to model multiple species may provide insights into the relative influence of climate change on populations trends (Johnson et al., 2017).

Conclusion

Our analysis shows a strong negative correlation between warm summer temperatures and Tufted Puffin nesting range, particularly along the California Current. Construction of SDMs utilizing two different emissions scenarios for the year 2050 show southern range contraction and suggest a high risk of Tufted Puffin extirpation in the California Current large marine ecosystem. Ensemble projections support preliminary analyses suggesting that temperature is driving the current puffin population declines and colony loss. SDM model results can provide valuable input for conservation decision processes. Specifically, our work provides the foundation for evaluating the threat of climate change and increased summer temperatures on Tufted Puffin breeding range.

Supplemental Information

Supplemental Information 1 Complied Tufted Puffin colony survey data

Click here for additional data file.

Supplemental Information 2 Code

Click here for additional data file.

Supplemental Information 3 Pearson et al., unpublished data on preliminary occupancy analysis

Click here for additional data file.

Supplemental Information 4 Hodum et al. unpublished data

Click here for additional data file.

Supplemental Information 5 Supplemental Materials

Click here for additional data file.

Figure S1 Projected past Tufted Puffin habitat along the California Current

Tufted Puffin breeding habitat range projection map exclusive to California Current (32°N–48.5°N). Probabilistic map, color bins display percent probability of grid cell representing suitable habitat under 1950 climate. Map tiles ©Stamen Design, underCC BY 3.0. Data ©OpenStreetMap, underODbL.

Click here for additional data file.

Table S1 Environmental variables, measurements and units

Click here for additional data file.

Table S2 Time periods corresponding to model biological and environmental inputs

Click here for additional data file.

We would like to thank Shawn Stephenson of the United States Fish and Wildlife Service, Laurie Wilson of Environment and Climate Change Canada and Robert Kaler of the Alaska Department of Fish and Game for their help in compiling Tufted Puffin survey data. We would also like to thank Eric Ward of the National Oceanic and Atmospheric Administration for his help in model parameterization.

Additional Information and Declarations

Competing Interests

Author Contributions

The authors declare there are no competing interests.

Christopher J. Hart conceived and designed the experiments, performed the experiments, analyzed the data, contributed reagents/materials/analysis tools, prepared figures and/or tables, authored or reviewed drafts of the paper, approved the final draft.

Ryan P. Kelly analyzed the data, contributed reagents/materials/analysis tools, prepared figures and/or tables, authored or reviewed drafts of the paper, approved the final draft.

Scott F. Pearson conceived and designed the experiments, contributed reagents/materials/analysis tools, authored or reviewed drafts of the paper, approved the final draft.

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
