# Peer review of "Will the California Current lose its nesting Tufted Puffins?"

_PeerJ, doi:10.7717/peerj.4519_

## Round 0.1 · original submission · Minor Revisions

Dear Christopher and co-authors,

While the three reviewers are very positive toward your paper, they have raised important points and suggestions that will significantly improve the manuscript. Please pay particular attention to Reviewer 1's comments on "validity of the findings". I share the reviewer's concern, among others, on why only 3 climate models are used when there are several of them in the CMIP archive. Please either add more models or provide a compelling justification with supporting references (e.g., on model performance) and how this choice may impact on the magnitude of projection uncertainties (using references that show projections of the California Current system, e.g., Brady et al. 2017, GRL, doi: 10.1002/2017GL072945, etc.) .

It is also important to indicate confidence level to statistical inferences where applicable (e.g., 95%, 90%; or state the p-value). For example, for the projections (L345-370), how confident are we on this statement: “..projects Tufted Puffins to lose 18% of their occupied range under RCP 4.5 and 25% under RCP 8.5 by 2050 (Table 3).”?

Based on the reviewers' comments as well as my own evaluation, this paper requires a relatively minor but a careful and mandatory revision.

Thank you again for submitting your article and I look forward to receiving a revised manuscript in due course.

Best regards,
Agus

·

Basic reporting

This manuscript is well-written and has a clear and timely conservation application. The introduction does a good job in laying out the climate change threats to the tufted puffin and the relevance of species distribution modeling for informing the Endangered Species Act listing decision and conservation actions in the face of population declines and colony losses.

The Figures and Tables are generally useful and well-described, with the following recommendations:
--Tables 1 and 2 could be converted to supplemental tables.
--In Figure 4, the green color used for a higher probability of occupancy is too difficult to distinguish from the green background shading of the map; I recommend changing this green to a different shade or making the continent a single color that can be distinguished from the legend colors. It’s also not clear why there are 4 occupancy categories in panel A and only 3 in panels B and C.
--In Figure 5, I recommend shading the histogram bars with stippling or striping rather than using solid colors so that the bars for the different model algorithms are easier to see.

In the Supplemental excel spreadsheet showing the data for the Tufted Puffin colonies, data from different sources are merged in a sloppy way, so that the data don’t match with the column headings. This should be fixed so that there is a single set of headings in the top row, and data throughout the spreadsheet conform with these headings, with blanks or N/A for cases where there are no data for that column heading in that dataset.

Experimental design

SDM is an important tool for assessing potential climate impacts on species distributions, and this manuscript provides an important application of this tool for the tufted puffin, which is suffering from significant declines in occupied colony sites.

SDM has well-known limitations that can influence the model results (see for example the discussion in Porfirio et al. 2014, Improving the use of species distribution models in conservation planning and management under climate change, PLoS ONE 9(11): e113749). The authors do a good job trying to account for some of these limitations, for example by using three different statistical models as well as an ensemble model, reporting two model performance metrics, and the selection of bioclimatic variables by eliminating variables with high collinearity as well as a basis in puffin ecology.

However, I recommend that the authors also justify the selection of the three GCMs that were chosen, particularly because GCMs can vary widely in precipitation projections, although less so in temperature projections. Do these GCMs represent a representative spectrum of future precipitation and temperature projections from the larger GCM model set?

In addition, using the ensemble mean of projected climate variables as the input into an SDM can be problematic, and it is recommended that multiple SDMs be run using a range of climate models as input, producing a range of species distribution maps which must then be combined (see Porfirio et al. 2014). One useful way to combine these maps can be to produce a map showing the agreement between the models—for example, pixels where all models agree a potential suitability above a threshold of 0.5. It’s not imperative that the authors do this exercise, but it’s something to consider.

It was unclear how you handled colonies that were extirpated during the current time period (1950-2008). Did you rank colonies that were extirpated during the current time period as absent?

I was glad to see that you estimated the correlation between SST and air temperature since studies of puffins (and related seabirds) indicate that SST has an important influence on nesting, typically through effects on prey. Your method appears to assume that SST in the 10km x 10km marine pixel nearest the colony site provides a good representation of SST in the puffin foraging range from that colony site, which, as you note, can extend to 40 km. However, I imagine a single pixel is a good enough representation of the marine foraging range, without the need to do additional analysis.

The methodological description of the PCA that you use to compare variance in environmental variables between areas where colonies collapsed in recent years to those that persist could use a bit more explanation. Although you say “recent years,” are you using the occupancy and climate data from the current period from 1950-2009?

Model calibration: You state that you constructed 20 models for each algorithm, but you should add more explanation of what the model structure for the 20 models looks like.

Validity of the findings

The results are interesting and informative as to the projected changes in nesting habitat suitability under rising temperatures, and are useful for informing conservation action for puffins.

There is a big difference in the percent range change projected for the California Current between the GLM compared to the GBM and RF approaches (-89 compared to -30). It would be helpful if you could comment on why there may be such a large difference in the results from these statistical methods.

There is also no difference in the ensemble range change projections for the California Current between the RCP 4.5 and 8.5 emissions scenarios, which is surprising and could also benefit from some proposed explanation.

You briefly acknowledge that that the range change modeling results show little agreement among model pixels in future suitable habitat. Because one of your stated purposes of the SDM exercise is to identify priority areas for conservation protection, it would be appropriate to add a note of caution since there appears to be little agreement in model results on the exact locations that are likely to remain suitable, although there is a trend of decreasing suitability in the southern portion of the range. In other words, it may be premature to deprioritize certain areas for conservation efforts given this uncertainty in model results. It might be worthwhile to show areas that had high agreement on future habitat suitability across models.

The moderate suitability of some colony sites in southern and central California under the RCP 4.5 and 8.5 projections is interesting, and I’m wondering what allowed those particular sites to stay somewhat suitable. It might be worthwhile to discuss the concept of climate refugia for those sites.

In terms of the potential for range expansion, it may be worth noting that puffins most often breed on offshore rocks and islands, as well as cliffs on the mainland, because of the absence of mammalian predators, and that predator presence provides an important limitation on suitable breeding habitat for puffins.

Additional comments

This manuscript makes an important contribution to understanding the potential consequences of climate change for tufted puffins, and has a timely and direct conservation application for this declining species.

I made some minor text edits in the attached manuscript.

·

Basic reporting

This study uses species distribution models to evaluate projected changes in suitable habitat for Tufted Puffins under 2 climate scenarios, and explores the potential drivers of population declines. Results suggest that most habitat in the California Current will become unsuitable for nesting Tufted Puffins, driven primarily by rising summer temperatures. This information will be extremely valuable for prioritizing long-term conservation actions, but also for regulatory processes that aim to protect species across their range.

I found this manuscript to be very well written making rather complex modeling procedures relatively easy to understand and evaluate. The introduction adequately describes the challenge of linking species declines to climate change, but also the importance of doing so for the purposes of both conservation planning and regulatory decision-making. Some specific comments on Tables, Figures and supplemental material follow (and see annotated pdf attached to this review).

If space is an issue, I think both Table 1 and Table 2 can be removed as they can easily be described in the text.

Figure 2 – I would like to see consistency in terminology used. In the figure title you use “occupied” and “unoccupied” but in the figure legend you use Absence and Presence. I suggest one or the other.

Figures 3 and 4 – I’m not sure whether this would add value to the Figures or not, but I would be interested in seeing a composite panel whereby you combine say Panel A (current projection) with Panel C (worst-case scenario) highlighting areas that decline in Probability of Occupancy (colour red) and those that increase in Probability of Occupancy (colour green?). Those areas that don’t change would be some neutral colour. I wonder if this type of figure would highlight better the specific areas in need of priority conservation actions?

Supplemental Material - Compiled Tufted Puffin colony survey data – file may be corrupt? I can’t seem to open this.

Experimental design

Research question was well defined and the results highly relevant to regulatory process related specifically to Tufted Puffin. Methods could also be applied to other species if quality and time frame of data were of similar quality as those of the Tufted Puffin. I do not have experience with computing SDMs myself, so cannot comment/review on the details of the analyses, however, the description of the methods was clear and on the surface sounded logical and robust. My only suggestion would be to add some of the text described in the supplemental material related to the ‘past model’ projection into the main manuscript as I was not clear of its relevance on first reading.

Validity of the findings

Conclusions related to range contractions in the south and potential expansion in the north are well presented and could have significant application to both short-term and long-term conservation actions. I would be interested to know whether there exists the detail to examine site-specific information. For example, the maps presented are at a scale where one would have difficulty identifying a specific colony that might be destined for extirpation. Although it gives a great overview what the species range might look like in another 20 years, could priorities be set based on the information presented here or would we need access to better spatial resolution? The findings also provide some evidence that temperatures in particular are driving the Tufted Puffin declines, although the relationship between temperature and puffin prey remains one worth increased attention.

Additional comments

This is a very well written paper that should be published with some minor edits. It will provide valuable information to regulatory agencies trying to assess the status of the species (both in the US and Canada), and may help direct colony-specific conservation actions in the future. Great work!

·

Basic reporting

This is a well-written manuscript overall. Following are comments focused on the background context provided in the Introduction and some more general editorial comments:

While I appreciate the concision of the opening paragraph, it would be helpful to broaden it slightly by including illustrative examples for the results of warming climates that you provide.

Line 58: Write out US Fish and Wildlife Service before providing the acronym.

Lines 82-84: It would be helpful to have the conservation actions specified for WA and Japan. State that it has been listed as Endangered at the state level. This is a stronger statement than how it is currently phrased.

Line 106 and 108: It should be “fledging” success unless you are specifically referring to the success of fledglings, which the Gjerdrum et al. paper does not emphasize. Does this need to be defined for a general scientific audience? And “Sea Surface Temperature” does not need to be capitalized.

Line 116: add the acronym “ESA” following your mention of the Endangered Species Act. The acronym appears later in the text but is not introduced previously.

Lines 143-144: I would suggest rewording the final phrase of the sentence to improve concision, as follows: “… can be used in both regulatory (e.g., Endangered Species Act listing decisions) and conservation planning contexts.”

Line 203: capitalize “puffin”.

In the Discussion, "auklet" needs to be capitalized in the common name of Cassin's Auklet.

This is a minor editorial point, but the word “data” is plural. The text should be edited to reflect that.

There is some inconsistency in tense in both the Methods and Results sections, moving between present and past tenses.

Experimental design

In general, the Methods section is well structured and thorough in its description of the modeling approach used in the study. Although the modeling presented in this study is beyond my expertise, the authors explain the process thoroughly and helpfully include justifications for various decisions they made in building the models run.

I have a couple of comments to offer for the Methods section:
Lines 151-152: It might be helpful to explain your process for determining factors relevant to Tufted Puffin breeding phenology. Someone who is not an expert on the species might really benefit from an expanded explanation here.

Lines 160-161: While I understand the idea behind the cropping process, using locations within 200 km of the ocean seems to be overly generous given that they are obligate coastal/island breeders. Shouldn’t the scale be more restrictive, possibly using the linear dimensions of your grid cell size of 10 km? If 200 km makes more sense from a model construction perspective, it would be helpful to have that explained explicitly.

Validity of the findings

The results are robust and presented in a clear and compelling manner. The figures are well-designed and contribute to both the Results and Discussion.

The major points in the Discussion are both compelling and appropriate and are completely consistent with the findings presented in the Results section.

The authors effectively frame the implications of their work not just in the context of the focal species but also as a general analytical approach applicable for many other species of conservation concern.

Additional comments

This is an impressive body of work represented in the manuscript. In addition to it being useful for researchers interested in state-of-the-art ways to model species responses to climate change, the results are particularly timely for conservation planning for the focal species, Tufted Puffins.

---

## Round 0.2 · accepted · Accept

Thank you for the revision. I am pleased to accept this article. Please double check throughout the manuscript to avoid errors in reporting, as well as typos (e.g., L226: “explanitory”; L441: “the represent”; L493: “south” should be “to the south” or “southward”? L499: “highlight” - for plural).

·

Basic reporting

I support the authors' revisions.

Experimental design

I support the authors' revisions.

Validity of the findings

I support the authors' revisions.

Additional comments

The authors did a thorough job responding to and addressing my concerns and suggestions, as well as those of other reviewers. I support the authors' revised analyses and edits which have made the study clearer and analytically more rigorous. I recommend this revised article for publication. It will make an important contribution to the scientific literature and to advancing conservation action for this threatened seabird.